# Perceptions of the Food Environment and Access among Predominantly Black Low-Income Residents of Rural Louisiana Communities

**DOI:** 10.3390/ijerph17155340

**Published:** 2020-07-24

**Authors:** Denise Holston, Jessica Stroope, Matthew Greene, Bailey Houghtaling

**Affiliations:** School of Nutrition and Food Sciences, Louisiana State University AgCenter, Baton Rouge, LA 70803, USA; jstroope@agcenter.lsu.edu (J.S.); mgreene@agcenter.lsu.edu (M.G.); BHoughtaling@agcenter.lsu.edu (B.H.)

**Keywords:** food insecurity, rural, food environment, low-income

## Abstract

Food insecurity in rural settings is complex and not fully understood, especially from the perspective of low-income and Black residents. The goal of this study was to use qualitative methods to better understand experiences with food access and perceptions of the food environment among low-income, predominately Black rural Louisiana residents in the United States. Data were collected from focus group discussions (FGD) and focus group intake forms. Study participants were all rural residents eligible to receive at least one nutrition assistance program. FGD questions focused on perceptions of the food environment, with an emphasis on food access. Participants (*n* = 44) were predominately Black and female. Over half (*n* = 25) reported running out of food before the end of the month. Major themes included: store choice, outshopping, methods of acquiring foods other than the grocery store, and food insecurity. Concerns around price, quality, and transportation emerged as factors negatively impacting food security. Understanding residents’ perceptions and experiences is necessary to inform contextually appropriate and feasible policy and practice interventions that address the physical environment and social conditions that shape the broader physical food environment in order to achieve equitable food access and food security.

## 1. Introduction

According to the U.S. Department of Agriculture (USDA), food insecurity is “the limited or uncertain availability of nutritionally adequate and safe foods or limited or uncertain ability to acquire acceptable foods in socially acceptable ways” [1]. The prevalence of food security in the U.S. differs by household location and sociodemographic characteristics and is a barrier for adherence to the 2015–2020 Dietary Guidelines for Americans. [2] Approximately 11.1% of U.S. households were estimated to be food insecure in 2018, with higher rates documented among households in rural (12.7%) and southern (12%) U.S. areas. Households with low income (29.1%), with children (13.9%), and those headed by Hispanic (16.2%) and non-Hispanic Black populations (21.2%) also experienced food insecurity at substantially higher rates than average [1].

Household food security and dietary quality is influenced by food access including the availability, affordability, accessibility, promotion, and acceptability of dietary options within built (e.g., stores/restaurants), wild (e.g., fishing access), and cultivated (e.g., community gardens) environments [3,4,5]. Food access barriers are also consistently documented among rural, southern, ethnic/racial minority, and low-income U.S. communities. Several federal nutrition programs including USDA’s Supplemental Nutrition Assistance Program (SNAP) and Special Supplemental Nutrition Assistance Program for Women, Infants, and Children (WIC) aim to improve household access to food by providing monthly benefits for food and beverage purchasing at authorized outlets [6]. However, while such programs are integral for improving food security and public health, [6] they are unsuited in isolation to solve complex issues of food and water access among vulnerable U.S. communities [7].

Improvements to economic, social, and environmental resources in rural and underserved communities are required to mitigate high rates of food insecurity and preventable noncommunicable diseases that are diet related, especially among rural non-Hispanic Black residents who experience the highest rates of mortality in the country [8]. In a critical literature review, Carter et al. found evidence suggesting residence in rural areas is protective against food insecurity mainly due to the social environment and unique factors at the local or “place” level reported within locations investigated [9]. There are also a number of studies [10,11,12,13,14] that find numerous food access and insecurity barriers among rural households that may be unique to rural settings in certain contexts. For example, households with lower income in rural areas may lack transportation or rely on social networks to access built, wild and cultivated community food options. There may be also be a false perception that rural areas have access to local agricultural products in abundance [11], although in many contexts these products are outsourced or largely not intended for human consumption [15]. Rural residents with low-income also use more of their total budget for food purchases than urban counterparts [16]. Additionally, according to some research, access to jobs and other opportunities that mitigate food insecurity is often based on one’s social and family network, particularly in the rural South [17].

Understanding the perceptions of rural residents from low-income communities about the local food environment and lived experiences with food access is needed to better understand food insecurity and inform contextually appropriate and feasible policy and practice interventions. This was the approach of the present study conducted among predominantly Black low-income residents living in rural Louisiana communities with high rates of obesity, poverty, and food insecurity [18,19,20].

## 2. Materials and Methods

This was a qualitative study guided by the constructivist research paradigm to gain an in-depth understanding of the lived experience of study participants [21]. The constructivist paradigm asserts that researchers are not objective observers. Rather, generated knowledge is co-constructed between researchers and study participants. A researcher’s past experience is seen as adding value to the research process, rather than introducing bias [22]. The lead author is an experienced qualitative researcher with seven years of experience working to improve local food environments in the rural communities targeted by this study. Research recruitment efforts were guided with input from local Cooperative Extension Service agents, who work closely with community members and local community organizations serving the population studied.

### 2.1. Setting

Louisiana has the 3rd highest food insecurity rate in the U.S. [20]. Focus groups were held in five rural Louisiana parishes (counties) with obesity, poverty, unemployment, and food insecurity rates higher than state averages. Parishes were classified as rural if they met the Rural-Urban Continuum Code (RUCC) classification (i.e., RUCC 4–9) [23]. Four of the five parishes studied sit in the Mississippi Delta region, an area with a long history of rural poverty and large percentage of non-Hispanic Black residents [24] (see Table 1).

### 2.2. Data Collection

All study procedures and documents were reviewed and approved by the Louisiana State University Agricultural Center’s Institutional Review Board (HE20-16). Purposeful sampling was used to recruit low-income participants eligible for federal nutrition assistance benefits including SNAP. Recruitment was conducted in coordination with parish Cooperative Extension Service. Agents leveraged community connections to recruit participants for focus group discussions (FGD) using word of mouth, social media, and fliers posted at community venues. Informed consent was provided by all participants and a demographic survey was completed prior to focus group commencement.

Using a flexible, semi-structured discussion guide, FGD facilitators asked participants how they perceived their food environment, how they acquired food, and about barriers and facilitators to accessing healthy foods. The discussion guide was developed according to rural food access literature [9,12,13,29] and prior work conducted in these rural parishes which indicated that the food environment was a barrier to healthy eating. Questions were also designed to facilitate discussion and sharing of knowledge among participants about resources available in their communities. Participants were encouraged to generate potential strategies to improve local food environments. Increasing research participants’ knowledge and moving participants to action are important criteria for “authenticity,” or quality, in constructivist qualitative research [30]. Participants were compensated for their time with a $40 check that was mailed to their address after the study. Six focus groups were conducted with a total of 44 participants. Each session was audio-recorded with permission. Focus groups continued until theoretical saturation was reached meaning additional FGD did not generate new insight [21].

### 2.3. Data Analysis

Data collection and analysis used an iterative process to determine the point of saturation. Focus groups were transcribed verbatim and transcripts were analyzed using Dedoose, a web-based qualitative research software which allows multiple researchers to independently code transcripts and compare among coders to reach inter-coder agreement [31]. Two members of the research team (M.G., J.S.) coded transcripts using this method and any disagreements were resolved through consensus and approved by the lead author.

Following the constructivist paradigm used, codes were derived inductively from participants’ responses, rather than deductive coding following a framework or codes determined a priori. Structural coding was used to segment transcripts into sections for analysis, and descriptive codes were used to label these sections according to the research questions to which they might apply [32]. Initial and in-vivo coding were then used to derive final codes from participants’ responses to interview questions.

## 3. Results

A total of 44 adults with low income residing in five rural parishes participated in this study. Participants were majority Black (*n* = 41) and female (*n* = 36) and on average 52 years old, ranging from 26 to 81 years. Employment status included full time (*n* = 9), part-time (*n* = 10), unemployed (*n* = 13), being on disability (*n* = 7) and seasonal work, which included field labor for day wages (*n* = 2). Almost all participant households received federal nutrition assistance program benefits (*n* = 43), with a lone participant eligible based on household characteristics, although non-participating. Thirty-two participants had at least one functional vehicle for their household. Participants reported high rates of food insecurity, with 25 reporting they had run out of food before the end of the month within the last year. Additionally, 19 participants reported there had been a time within the last 12 months when they did not have enough food to feed themselves or their family. Participants indicated price as the most important factor when shopping, followed by product quality (see Table 2).

Major themes that emerged from FGD included: store choice, outshopping, methods of acquiring foods other than the grocery store, and food insecurity, with price, quality, and transportation interacting with each theme. Participants also shared potential solutions to food access issues in their community, as prompted by the discussion guide.

### 3.1. Store Choice

When asked about what motivated store choice decisions, price and quality stood out as primary deciding factors for participants from every focus group, with price being the most important deciding factor for almost all participants. Participants often searched for best prices or compared sales using grocery fliers, for example, “We compare these prices. We get the sale paper and we compare the price. And we also go for the cheapest.” Participants were keenly aware of small differences in price. “It could be a penny difference, I’m still going there (laughs). Things add up.”

Quality was often mentioned in the same sentence as price and was also a top priority in all focus groups. Participants complained that produce, meat, and other products were not fresh in the local store, saying “the fruits and vegetables that are at the grocery store here are overpriced and 9 times out of 10 they’re already spoiled” and “if you check the expiration date, they was like from way last year. They still got it out there.” Issues with quality and price frequently forced residents to leave their parish for better options.

#### 3.1.1. Outshopping

Outshopping, or having to leave the parish to find lower prices and better quality, was a major theme in every focus group. Participants from three focus groups close to neighboring states mentioned crossing state lines to find better options. Most participants reported traveling at least 30 min to acquire food. While some participants had their own vehicle, the need to pay for out of town rides came up in every focus group. Most participants said they paid between $10–40, even if with a family member, saying, “Ain’t nothing free.” Some participants reported paying for the ride in cash, while others said they filled up the ride’s gas tank. In one parish, “selling coupons” (a reference to allowing someone else to use your SNAP benefits) came up as a way that people pay for a ride to purchase groceries, saying, “that’s when selling them coupons can come in handy.” Paying for rides was viewed as a mutually beneficial arrangement, framed by the following participant, “it be money issues coming back and forth. It’s gas. So it saves both of us pretty much.” When asked how frequently they would pay for rides, participants indicated that they tried to only pay for one ride per month, soon after benefits were received.

In all focus groups, trips to local grocery stores were generally only to pick up very few items, or motivated by a big sale. When asked about shopping at the grocery store in town, one participant responded, “With what they’re charging it’s not worth it.” Participants directly connected outshopping with the need to make their benefits last. When asked about why they left town to shop, one participant replied,


*“I get food stamps. I’m not ashamed of that. I’m proud of that. That’s a blessing … But the thing of it is, I be trying to stretch it, you know. That’s what I be trying to do is stretch it. If I go to [local store], I ain’t get nothing on that. And the meats are so high there … I find those meats and things cheaper when I go [out of parish] than I do at these stores around here.”*


#### 3.1.2. SNAP/WIC

In every focus group, almost all participants received SNAP benefits, though many participants mentioned receiving only $16 or $35 per month. One participant expressed her frustration, “I can’t go nowhere with sixteen.” Another one shared, “The benefits, they don’t fit. They don’t give enough. It’s not enough and if I went and made $100 more a month, they’re gonna take half my stamps away … The more you make the more they take. And then you’re paying cash, and cash is a whole lot more than stamps because you don’t have to pay taxes on stamps.”

Participants were very aware of the need to make their food dollars go as far as possible, saying, “I try to stretch the little food stamps I get.” A fellow participant responded, “I still don’t know how you stretch $35.”

Female participants in households with young children also discussed reliance on WIC. Differences from store to store in WIC approved items caused frustration, especially when stores did not carry an item that was a part of their WIC package, causing the women to visit multiple stores to be able to get all of their items. With limited transportation and often paying others for rides, this was a considerable burden on the WIC recipients. One mother explained,


*“Certain items they out of at this store, you can’t get your WIC at all. You gotta go to a whole different store to cash that WIC. And now you got to like, pinpoint where do I got to go to get such and such. And it’s crazy.”*


#### 3.1.3. Price Gouging

Participants among 4 focus groups explained prices consistently decreasing during the last half of each month, after SNAP benefits are exhausted. In Louisiana, SNAP benefits are provided between the 1st and 15th of every month [33] “Beginning of the month it be high. All the food be extra high. But like close to the end of the month, that’s when the prices wanna go down.” Another shared, “After you done finished your stamp that’s when the best sales come.” Participants expressed shared frustration at this perceived practice and felt that residents who rely on SNAP were excluded from being able to take advantage of the best sales, which they reported almost always came at the end of the month.

#### 3.1.4. Active Transportation to Grocery Stores

Participants were asked if they walked or biked to go get groceries. Participants from all but one focus group felt that it was feasible for some to walk or bike to get groceries. Individuals from 3 parishes shared that they walk or bike to the grocery store when needed, saying “I walk sometimes” and “I ride a bike every now and then.” Only male participants mentioned biking. Heat, distance, and health issues were mentioned as barriers to active transportation.

### 3.2. Ways of Acquiring Food Other Than the Grocery Store

All focus groups reported using a variety of built, cultivated, and wild environments to acquire food.

#### 3.2.1. Convenience Stores/Dollar Stores

No participants viewed convenience stores as a viable option for acquiring food. When asked if they wished more local corner stores or convenience stores accepted SNAP, the response was unanimous, illustrated by one participant: “Lord, no.”

Dollar stores, however, served an important role for many participants, particularly for “sides and stuff.” Participants reported using dollar stores to purchase non-perishable items, saying, “I would have just enough money to make it there. You go into [local grocery store] and … you can’t get anything.” Dollar stores were not specifically asked about in the focus group interview questions, but were mentioned by five of the six focus groups—the only group that did not mention dollar stores was also the only FGD that occurred in a community with a Walmart.

#### 3.2.2. Food Pantries/Commodity Boxes

In each focus group there were participants who were more aware of the local food resources than others. When asked if participants were able to use food pantries or food banks, participants in the same focus group responded, “I’ve never used one. I didn’t think [town name] had one”, and, “if I had to go for a food bank, it’d be in Monroe” [30 min away]. To this, a third participant responded, “There is one down here too. You go to the church down on 10th street.” This type of dialogue, where some focus group participants informed others of existing resources in the community, happened in three focus groups.

One participant mentioned occasional truckloads of food donations available to the community. When asked how people found out about opportunities like this, responses included: “Facebook”, “somebody you know”, and “somebody told somebody told somebody told somebody.”

Many participants also participated in the Louisiana Commodity Supplemental Food Program (referred to by participants as “commodities” or “commodity boxes”), available to low-income persons over 60 years old. Recipients expressed frustration that the boxes often included pinto and black beans (instead of red beans or black-eyed peas). In one focus group, a participant said, “the commodities … help some of the people, but then everybody don’t use that pinto beans, black beans, you know I mean that’s just …” The group responded with “ugh” and multiple groans. Specific dislike of black beans was voiced by other participants as well. But appreciation at the variety included was also expressed: “because they do change it up, you know, sometimes you get rice, sometimes you get grits. It’s a variety of stuff. And they give the salmon in the can which is good. And that’s healthy. Veggies. That’s healthy.” Another participant added, “I love the fruits in the can! And my oatmeal, oh Lord!”

#### 3.2.3. Bartering

When asked about trading or bartering for food, five of the six focus group participants responded positively, with the exception of a few individual participants saying they lived too far from neighbors to be able to do this. One focus group, located in the most populous parish studied (population 25,396), found the question puzzling, saying, “That was just in the olden days.” Others found this an essential part of making it through.

#### 3.2.4. Gardening

Questions about gardening yielded mostly negative responses, with a few outliers. Gardening was viewed as impractical because of heat, time constraints, past garden failures, and startup costs. Comments included: “Don’t nobody have no garden now” and “I tried to plant tomatoes, don’t ever work.” Individuals in 4 of the focus groups mentioned produce from gardens as a source of food for their household. One participant found gardening to be an important contribution to her food supply, saying, “I live by myself, though I support my mother and cook for her. A small raised garden bed goes a long way in providing fresh food. Not the whole year, but a good part of it.”

#### 3.2.5. Hunting/Fishing/Gathering

Participants from five out of the six focus groups indicated that fishing and hunting were a part of how their households acquired food, including alligator, catfish, crawfish, deer, and sac-au-lait (white perch). The levee along the Mississippi River (adjacent to four of the six focus group parishes) was mentioned by several participants as a popular location for fishing. Participants indicated they fish for fun and for food. One participant shared, “there are times that we may not have funds for what we want [to eat], but we get by. Anytime I run out, I go and catch me a fish”. Pecan trees, whether on personal property or public land, were mentioned as a food source in four focus groups. A few participants mentioned selling extra pecans for cash to pecan processors.

### 3.3. Food Insecurity

Beyond conversation about ways in which to acquire food, the focus groups also discussed food insecurity. Most participants said they had to adjust how they fed their household to be able to make it to the end of the month by changing what they eat. Some specifically mentioned the need to fill up their children with starchy foods like rice and potatoes so that they could afford to feed their family. One participant shared that by the end of the month, “we’ve had to alter choices. Less quality.” Another spoke of limited choices, saying, “when my stamps run out and I’m on a fixed income—my money runs out and I look in the cupboard and I have a can of beans or something.”

Very few participants indicated they go hungry, but when asked if they sometimes struggle to feed their families, participants from each focus group responded yes. Wider family and social relationships helped bridge the gap between the end of one month’s benefits and cash, and the days or weeks before a trip to the store could be afforded. The importance of family networks was emphasized, with participants saying, “everybody here came out of big families, so we know how to get by” and “if you’re an elderly person, you have to have some kind of family, a sister or daughter or something. You have to have help”. Other social networks were mentioned. One participant shared, “our church family helps us”. To this another participant replied that her church does not help.

Friends and family were mentioned as the source for information about access to food, such as finding out about occasional food donation events. The ability to feed their families healthy foods was limited, however. One mother shared,


*“To feed somebody healthy … it costs way more. A lot of people are pretty much looking at food that’s filling. You know what I’m saying? So you don’t have to eat as much. Like rices and starches and stuff like that to make sure food can go further, I guess. But healthy food, it really costs.”*


Another shared, “pork isn’t really good for us, but we eat it anyways because it’s cheaper. Could get some fish, salmon, or chicken or turkey, but it’s high.” Cost was perceived as a barrier to providing healthy foods for their families by participants in every focus group.

### 3.4. Potential Solutions

Participants were asked for their thoughts on what potential solutions might be. Focus group discussion facilitators asked what participants would do if they were in charge.

#### 3.4.1. Water

Residents from two parishes with ongoing water issues, including frequent boil orders and occasional shut-offs, [34,35,36] explained they did not trust the safety of tap water even when local officials declare it is safe to drink because it is often brown with sediment, saying “you can’t ever drink that” and “there’s sand in the bottom of it.” Residents from these two parishes shared that fixing the water problems would allow them to spend more money on food, as they would no longer have to purchase bottled water.

#### 3.4.2. Gardening

Gardening came up again when the focus group participants discussed possible solutions to improve food accessibility. Gardening was supported as a partial solution by some participants, but noted as unrealistic by others. One participant mentioned wanting to garden, but not being able to afford the materials, saying, “I don’t know about anybody else, but money’s tight for me. Buying the materials to build a raised garden. I have a bad back, so a raised garden would be the only one I could do. I’ve always wanted one, but I can’t afford the materials to build one.” Others retorted, “when that sun gets hot and those butter beans and field beans need pickin, it ain’t gonna happen.” The discussion leader asked specifically about community gardens, but this was not considered a viable option. Participants believed community gardens would end up being controlled by certain people in the community, without access for all. Three participants had the following exchange when asked if a community garden would help:

Person A: *“No. Certain people would try to take over. And get what they want out of it.”*

Person B: *“A lot of things around here are based on relationship. Who you know.”*

Person C: *“I was just gonna say that. If you don’t know the right people—and you can’t get in contact—if you’re not cool with the right people, then you’re still gonna be left out.”*

Though gardening (whether personal or community-based) was viewed as something that could help individuals, it was met with skepticism as a solution that would improve the parish food environments as a whole.

#### 3.4.3. Increase Competition

Participants from two parishes suggested that a lack of competition contributed to high prices and expressed desire for more stores to open. One person shared the community needed “a little bit more competition so their prices come down.”

#### 3.4.4. Jobs

Ultimately, participants thought the most impactful way to improve their food environment would be to provide job opportunities with reliable hours and a livable wage. Knowing the right people was viewed as the most important factor in being able to secure a job. Participants shared “you gotta know somebody, you gotta be kin to certain people to get a job down here” and “it’s ridiculous. Most of the jobs here they’d rather give it to a buddy or a son.” Even participants with prior experience outside parish reported being unable to secure jobs, attributing that to not know the right people. “It’s who you know around over here.”

## 4. Discussion

Food insecurity in rural settings is complex and not fully understood, especially from the perspective and experiences of low-income and Black residents [12]. In order to address factors related to food insecurity in rural settings, we must first document and describe food access among those who experience food insecurity. The goal of this study was to use qualitative methods to better understand experiences with food access and perceptions of the food environment among low-income, predominately Black rural residents. Exploring low-income resident perceptions and experiences allows researchers to better understand pathways for food insecurity and prioritize feasible, appropriate solutions and interventions.

In this study, we found that low-income and majority Black rural residents in Louisiana do not have adequate access to food in their community and experience barriers when acquiring food within and outside of the community. Study participants selected food stores based on food prices and perceived food quality. This finding is consistent with others in rural settings [13]. Further, a review by Carter et al. suggested that living in a rural area was protective against food insecurity [9]. In contrast, our findings suggest that living in rural locations, being poor and a minority may contribute to food insecurity. Previous research among rural food insecure persons has largely focused on White audiences, [12,29,37] and very few have included the perceptions of rural, low-income Black audiences, which experience a higher prevalence of food insecurity [11,14].

Participants typically only had access to one grocery store in their parish and chose not to shop at that store due to perceived high food prices and poor food quality. Participants thought these issues were due to a lack of competition among local food outlets in the parish. These perceptions of the local food environment have important consequences. For example, research by Garasky et al. demonstrated that perception of high local food prices and few food stores were associated with being food insecure [10]. Participants in our study reported leaving the parish they lived in to look for lower priced, higher quality food. Recent research has uncovered similar findings, demonstrating that high local food prices are extremely problematic for those on limited incomes [12,38] and compel rural residents to mainly shop outside of the local community e.g., ‘outshopping’ [10,12,13].

The practice of outshopping incurs costs in terms of time and transportation, with participants paying $10 to $40 for a ride to a store about 30 to 60 min away. Participants in our study reported that paying for transportation in the form of gasoline, money and/or food stamps decreased the amount of funds families had available for food each month. However, participants still thought this was a more economically feasible choice than paying high prices in their local communities [39,40]. High prices at the local store likely limited active transportation for groceries, as many participants felt outpriced to shop locally. Participants relied on car transportation and almost all indicated they did not regularly walk or bike to get groceries due to the distance, heat, and personal health limitations. Our findings, and others suggest that transportation was not readily available (e.g., due to fuel costs), even if participants reported having a vehicle in the household [40].

Responses from both the FGD and participant survey suggested that most participants ran out of SNAP or WIC funds before the end of the month even with outshopping, but the deficit was more severe if foods were purchased locally. Interestingly, Cafer and Kaiser found that SNAP significantly improves purchasing power of rural residents when compared to urban [41], but our results indicate that high local prices and limited access to transportation lessened the potential benefit of SNAP and WIC in the communities we studied. While federal nutrition safety net programs alone are not enough to reduce food insecurity [10], social conditions within a community, specifically a rural community, can further lessen the effect of these programs [40]. We found evidence of that in this study.

Another important finding from our study was rural residents’ perception that food prices and store sales fluctuated with timing of SNAP benefit distribution, with higher prices occurring early in the month to coincide with distribution of benefits. In Louisiana, benefit distribution occurs during the first half of the month on a rolling basis based on the last digit of a person’s social security number, with the exception of disabled and elderly recipients, who receive benefits during the first four days of the month [33]. More research is required to investigate if price fluctuations are related to SNAP issuance periods. A small body of work has identified price and promotional differences for sugar-sweetened beverages by low-income and SNAP status [42,43,44,45]. If and how these practices extend to other healthy and unhealthy products in rural, low income areas is unknown. Our findings contradict those found in a rural, agricultural community in California, where rural residents reported ample access to fruit and vegetables through farmer direct sales, road-side stands or farmers markets. While participants in our study represented rural agriculture communities, farmers in our study area typically grow soybeans and other commodity crops, with little production of fruits and vegetables. Additionally, the represented parishes have the highest number of farming acreage combined with the lowest total number of growers—there are few farms, but they are typically huge operations [15]. Our participants did not frequent farmers market or road side stands possibly due lack of availability, transportation and/or SNAP or WIC vouchers not being accepted.

The social environment can influence food access, but this interaction is not well understood [39,40]. One study suggests that the physical and social environment can place the most significant challenge to food access in rural areas [39] while another suggested that the social environment within rural communitive can be protective against food insecurity [9]. Our findings support this notion: the social environment, particularly social support networks within a rural community can influence food access, especially among marginalized populations [39]. Family, church, and other support networks helped some participants when they were running low on food, funds, and in some cases transportation. Benefits of support and family networks have been reported in rural literature [10,39] and highlights the dire situation of those who do not have access to a support network.

When participants suggested solutions to food access, comments about solutions were frequently connected to social capital and social support. For example, in several of the FGD, improved finances by way of jobs or employment were cited as a solution to improve food access. As this concept was suggested it was generally countered by other participants that jobs were hard to obtain and acquiring jobs was almost always based on “who you know.” In another example, community gardens were mentioned as a solution and was again countered with concerns that only certain people or groups of people would be able to benefit from the garden. Lastly, some participants only learned about a local food pantry at the FGD, which further demonstrates that rural residents who are not socially connected may have additional challenges with food access and food insecurity. Meaningful solutions to food insecurity must take into account the role of the social environment, social support networks, and social isolation.

To bridge the gap in food access, our study participants reported bartering for food, gardening, hunting/fishing, visiting food pantries, and shopping at dollar stores. Our participants generally had a negative opinion of home gardening, which supports a recent study reporting that rural food insecure persons were less likely to have a garden than food secure persons [14]. Other recent studies suggest gardening is perceived positively by white rural residents [12,39]. and may fill food access gaps [29]. This contrast in perceptions of home gardening in rural communities should be further explored to determine if gardens are a feasible way to improve rural food environments. Access to local rivers, lakes, and land gave participants other opportunities to acquire food through hunting and fishing. As many participants mentioned the difficulty of purchasing affordable lean protein options locally, policies to formally allow fishing access along rivers along with educational programs to share local fishing knowledge could help improve the food environment.

Despite the variety of strategies mentioned to supplement stretched food budgets, many participants did not know about available community food resources prior to participation in our FGD. Improving communication efforts around these resources and leveraging social networks may improve food access. Rural communities are not homogenous, and the variety of food acquisition strategies sources beyond shopping at grocery stores highlights their distinct dynamics. One singular strategy to improve the food environment is not likely to equitably benefit all rural communities, though a focus on promoting existing resources would likely help many food insecure households.

Data collection in this study occurred in January and February of 2020, shortly before the known COVID-19 outbreak. Early data from Urban Footprint suggested that one in three Louisianans now lives in a food insecure community [46]. As a result of the pandemic and the Families First Coronavirus Response Act (FFCRA), [47] food stamp allocations were distributed at the maximum allowance for four months in Louisiana (as of June 2020), which could have amounted to a 40% increase in monthly benefits [48]. Participants in our study indicated that they ran out of money or food stamps; we wonder if the increases in allocation and stimulus funding changed food access perceptions and experiences. At the same time, stimulus funds caused some people to lose eligibility for food stamps, causing some to become food insecure if they were not already food insecure [49].

### 4.1. Limitations

Participants were low-income volunteers recruited through the local Cooperative Extension agents and self-selection may have resulted in a non-representative sample. Due to the small size and close-knit nature of many rural communities, participants could have influenced group dynamics and each others’ responses. Lastly, participants’ experiences with food access may not be transferable to all low-income individuals residing in rural areas, especially those outside of the South. Despite limitations, this study fills a gap in research on rural food environments and Black residents with low income in the South—an important contribution given elevated rates of food insecurity in the region. Additionally, this study adds to our understanding of how the social and built environment interacts with food access and food insecurity, particularly in low-income, rural and majority Black communities. It also sheds new light on the rural food environment, rural food access, and rural food insecurity.

### 4.2. Public Health Implications

The rural food environment can present challenges for low-income residents, which may lead to food insecurity. Our findings indicate that the broader physical food environment and the social conditions need to be addressed to promote equitable food access and food security among vulnerable populations. Understanding resident perceptions and experiences can better equip researchers and communities to prioritize feasible interventions to improve food systems and food access.

Further research should examine the feasibility of rural transit and related funding opportunities to address the transportation barriers identified. Because rural transit may be a long-term project, communities may want to consider local, innovative rural transit solutions. Additionally, exploring how to promote existing community food resources should be a component of food access interventions in underserved communities. More research on how social networks and social isolation impact food security in rural areas is warranted. Finally, further research on examining ways to build social capital among low-income rural residents, especially low-income Black residents would be fruitful.

## 5. Conclusions

Rural food insecurity across communities is disparate and multi-layered. In rural Louisiana communities among low-income predominantly Black residents, we found that price drove outshopping behaviors, and that outshopping often depended on paying others for transportation and limited food dollars. The social and natural environments made food insecurity less severe for some participants, while others were unaware of existing community resources. Options for local purchasing were described as limited, costly and low quality, with many healthier items unaffordable or unavailable.

## Figures and Tables

**Table 1 ijerph-17-05340-t001:** Descriptive Characteristics of the Target Parishes.

	Louisiana Parishes					
	Assumption	East Carroll	Madison	Morehouse	Tensas	LA Average
Population, *n* ^a^	21,891	6861	10,951	24,874	4334	-
Black, % ^a^	29.4	68	62.4	48	54	32.7
Obesity prevalence, % ^b^	34.8	42.1	41.8	39.4	34.8	34.5
Poverty, % ^c^	17.4	48.6	37.8	28.5	40.0	19.4
Unemployment Rate, % ^d^	6.1	10.6	8.1	8.3	8.3	4.8
Food Insecurity, % ^e^	15	33	26	22	26	17
Food Pantries ^f^	4	1	1	4	3	-
Summer Feeding Sites ^g^	1	1	1	1	1	-

^a^ Source: 2019 Census Population Estimates. [25]. ^b^ Source: 2017 Louisiana Department of Health [26]. ^c^ Source: 2018 American Community Survey 5 year estimates [19]. ^d^ Source: 2019 U.S. Bureau of Labor Statistics Labor Force Data by County, 2019 Annual Averages [27]. ^e^ Source: Feeding America, 2019 [20]. ^f^ Source: LSU AgCenter inventory of food pantries in targeted parishes [28]. ^g^ Source: USDA Food and Nutrition Service Find Meals for Kids Tool [23].

**Table 2 ijerph-17-05340-t002:** Focus Group Participant Survey.

	Number
Number of Participants	44
Gender	
Male	8
Female	36
Race and Ethnicity	
American Indian	1
Black	41
White	2
Employment Status	
Full-time	9
Part-time	10
Seasonally employed	2
On disability	7
Unemployed	13
No response	3
Does your household receive benefits?	
YES	43
NO, not eligible	0
NO, eligible to receive	1
Have you run out of food before the end of the month in the last twelve months?	
YES	25
NO	19
Has there been a time in the last 12 months when you have not had enough food to feed yourself or your family?	
YES	19
NO	24
No response	1
Have you used a food bank in the last twelve months?	
YES	20
NO	24
How many working vehicles are there in your household?	
0	9
1	23
2	8
3	1
No response	3
What is the most important thing you look for when shopping for food? (Note: Some participants marked more than one response).	
Price	32
Quality	13
Location of store	0
EBT/WIC acceptance	11

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
