# Peer review of "Perceptions of the Food Environment and Access among Predominantly Black Low-Income Residents of Rural Louisiana Communities"

_ijerph, 2020, doi:10.3390/ijerph17155340_

Round 1
Reviewer 1 Report
I appreciated the opportunity to read this interesting and important paper. Overall the paper provides a useful contribution to research on rural experiences of food access and a good basis for further research and intervention. I would like to see a little more grounding in the literature on food environments (aware that a lot of this has focused on urban environments) and a slightly clearer framing in terms of what access means. One overall point is that I would also avoid the term 'subject' which seems out of keeping with the way we would talk about research participants.
Intro:
There is a large literature that speaks to individuals experiences and engagement with food environments. I know that much of this is based on urban settings but I think the paper would be strengthened by a stronger framing in terms of on experiences of food access rather than just causal factors.
Additionally you could explain a little more clearly exactly what you mean by food access as this means different things in different pieces of research (e.g., Turner et al 2018 use it to talk about physical access but not affordability, which they term affordability).
You raise obesity in the introduction as relevant in terms of food access. This is certainly true, but feels a bit thrown in so you may want to consider how you integrate mention of public health concerns.
(Applies to the discussion as well) I would appreciate a comment on why Carter thinks rurality is a protective factor for food insecurity. Without an explanation it seems like a straw man to knock down.
Methods
The description of coding was slightly confusing as it sounds a bit like you were using the research questions as codes, although you also say you were coding inductively. Please clarify.
You do not mention ethical approval. Please provide details.
I appreciate the moving to action element and desire to provide participants with useful resources, but the language of increasing participants knowledge feels at odds with a study that is trying to understand participant perspectives without imposing researcher perspectives.
Author Response
Thank you for reviewing our manuscript.
I appreciated the opportunity to read this interesting and important paper. Overall the paper provides a useful contribution to research on rural experiences of food access and a good basis for further research and intervention. I would like to see a little more grounding in the literature on food environments (aware that a lot of this has focused on urban environments) and a slightly clearer framing in terms of what access means. One overall point is that I would also avoid the term 'subject' which seems out of keeping with the way we would talk about research participants.
Thank you for pointing this out. Avoiding this kind of language is important. We have removed the two instances of the word “subject.”
Intro:
There is a large literature that speaks to individuals experiences and engagement with food environments. I know that much of this is based on urban settings but I think the paper would be strengthened by a stronger framing in terms of on experiences of food access rather than just causal factors.
We have added language to more explicitly connect the perspectives of the participants to their lived experience of food access. See lines 63-65 and 345.
Additionally you could explain a little more clearly exactly what you mean by food access as this means different things in different pieces of research (e.g., Turner et al 2018 use it to talk about physical access but not affordability, which they term affordability).
We revised a sentence in the introduction to clarify our definition of food access.
“Household food security and dietary quality is influenced by food access including the availability, affordability, accessibility, promotion, and acceptability of dietary options within built (e.g., stores/restaurants), wild (e.g., fishing access), and cultivated (e.g., community gardens) environments.3–5”
You raise obesity in the introduction as relevant in terms of food access. This is certainly true, but feels a bit thrown in so you may want to consider how you integrate mention of public health concerns.
This is helpful—we removed the reference to obesity (line 47-49). All though obesity is certainly important, it is not the focus of this manuscript.
(Applies to the discussion as well) I would appreciate a comment on why Carter thinks rurality is a protective factor for food insecurity. Without an explanation it seems like a straw man to knock down.
Thank you. We have added further explanation in the introduction (line 51-53) and discussion (line 405-406).
Methods
The description of coding was slightly confusing as it sounds a bit like you were using the research questions as codes, although you also say you were coding inductively. Please clarify.
Thank you for bringing this to our attention. We see how the description of our coding is unclear. Descriptive coding was used merely to label sections of the focus group transcript according to the research questions to which those sections might apply. The final codes used for our analysis were initial and in-vivo codes derived from participants’ responses. We have revised the manuscript to reflect this.
You do not mention ethical approval. Please provide details.
Thank you for bringing this to our attention. This information has been added to the manuscript. “All study procedures and documents were reviewed and approved by the Louisiana State University Agricultural Center’s Institutional Review Board (HE20-16).”
I appreciate the moving to action element and desire to provide participants with useful resources, but the language of increasing participants knowledge feels at odds with a study that is trying to understand participant perspectives without imposing researcher perspectives.
Thank you for bringing this to our attention. We should have specified that participants’ knowledge increased through group sharing of available resources, not researcher-led instruction on these resources. The manuscript has been changed to reflect this.
Reviewer 2 Report
This is an excellent paper on an important topic. Moreover the sample included in the study is very marginalized and, further, not as well researched as they should be. Their voices need to be heard. The themes are sometimes similar to what we have learned about people who are food insecure, but new themes have arisen due to this context. I found this paper to be an interesting read.
I like the constructivist approach and the fact that some questions were designed to guide participants to services. it is very appropriate to alert participants to services that can benefit them. I would have liked to learn a little more about how these services were presented to them.
Re methods: This seems like a very ethical study, but i couldn't find anything about an ethics board approval? I believe that this should be included in the write-up.
The issue of 'who you know' really bothers me. Do the authors have any ideas of how this can be further researched?
Author Response
Thank you for reviewing our manuscript.
This is an excellent paper on an important topic. Moreover the sample included in the study is very marginalized and, further, not as well researched as they should be. Their voices need to be heard. The themes are sometimes similar to what we have learned about people who are food insecure, but new themes have arisen due to this context. I found this paper to be an interesting read.
Thank you.
I like the constructivist approach and the fact that some questions were designed to guide participants to services. it is very appropriate to alert participants to services that can benefit them. I would have liked to learn a little more about how these services were presented to them.
Thank you for your comment. Participants learned about available resources through group discussion, rather than being informed about those resources by the researchers. We have edited the manuscript to clarify this.
Re methods: This seems like a very ethical study, but i couldn't find anything about an ethics board approval? I believe that this should be included in the write-up.
Thank you for bringing this to our attention. This information has been added to the manuscript. “All study procedures and documents were reviewed and approved by the Louisiana State University Agricultural Center’s Institutional Review Board (HE20-16).”
The issue of 'who you know' really bothers me. Do the authors have any ideas of how this can be further researched?
We think that further research on examining ways to build social capital among low-income rural residents, especially low-income Black residents would be fruitful. We included the following sentence, “Finally, further research on examining ways to build social capital among low-income rural residents, especially low-income Black residents would be fruitful.”